# Vertical structure of biomass burning aerosol transported over the southeast Atlantic Ocean

Harshvardhan Harshvardhan[1], Richard Ferrare[2], Sharon Burton[2], Johnathan Hair[2], Chris Hostetler[2], David Harper[2], Anthony Cook[2], Marta Fenn[3], Amy Jo Scarino[3], Eduard Chemyakin[3], Detlef Müller[4]

[1]Purdue University, West Lafayette, IN, United States
[2]NASA Langley Research Center, Hampton, VA, United States
[3]Science Systems and Applications, Inc./NASA Langley Research Center, Hampton, VA, United States
[4]University of Hertfordshire, Hatfield, Hertfordshire, United Kingdom

*Correspondence to:* H. Harshvardhan (harshvar@purdue.edu)

**Abstract.** Biomass burning in southwestern Africa produces smoke plumes that are transported over the Atlantic Ocean and overlie vast regions of stratocumulus clouds. This aerosol layer contributes to direct and indirect radiative forcing of the atmosphere in this region, particularly during the months of August, September, and October. There was a multi-year international campaign to study this aerosol and its interactions with clouds. Here we report on the evolution of aerosol distributions and properties as measured by the airborne high spectral resolution lidar (HSRL-2) during the ORACLES (Observations of Aerosols above Clouds and their intEractionS) campaign in September 2016. The NASA Langley HSRL-2 instrument was flown on the NASA ER-2 aircraft for several days in September 2016. Data were aggregated at two pairs of 2°×2° grid boxes to examine the evolution of the vertical profile of aerosol properties during transport over the ocean. Results showed that the structure of the profile of aerosol extinction and microphysical properties is maintained over a one to two-day time scale. In the 3-5 km altitude range, 95% of the aerosol extinction was contributed by particles in the 0.05-0.50 µm radius size range, with the aerosol in this size range having an average effective radius of 0.16 µm. This indicates that there is essentially no scavenging or dry deposition at these altitudes. Moreover, there is very little day to day variation in these properties, such that time sampling as happens in such campaigns, may be representative of longer periods such as monthly means. Below 3 km there is considerable mixing with larger aerosol, most likely continental source near land. Furthermore, these measurements indicated that there was often a distinct gap between the bottom of the aerosol layer and cloud tops at the selected locations as evidenced by a layer of several hundred meters that contained relatively low aerosol extinction values above the clouds.

## 1 Introduction

Aerosols are often considered as the most confounding element in the climate system when simulating parameters of the Earth's current climate. Their interaction with clouds makes the problem extremely complicated. The general topic of aerosol-cloud interaction has been of great interest in the scientific community: to quote the report of the Intergovernmental Panel on Climate Change (IPCC AR5) "Clouds and aerosols continue to contribute the largest uncertainty to estimates and interpretations of the Earth's changing energy budget*"* (Boucher et al., 2013).

In the context of these interactions, the interplay of biomass burning (BB) aerosol and the stratocumulus clouds in
the Southeast (SE) Atlantic is unique and crucial to the estimates of the energy budget of the region. This BB aerosol
arises from the seasonal burning (July-October) of agricultural residue in the southwestern African Savannah and
traverses large distances westward over the SE Atlantic Ocean. Unlike the aerosol from industrial activity and biofuels
that intermingle with clouds in many regions (Ramanathan et al., 2001; Mechoso et al., 2013), these optically thick
BB aerosol layers overlay vast stretches of marine stratus cloud in the SE Atlantic (Chand et al., 2009; Wilcox, 2010;
Adebiyi et al., 2015) where they have a direct radiative effect. The BB aerosol can also act as nuclei for cloud droplets
and so cause a potentially significant cloud albedo effect. Observations and modelling studies of such interactions in
the Southeast Atlantic and southern Africa regions include Diamond et al. (2018), Kacarab et al. (2020), Mallet et al.
(2020) and Gupta et al. (2022). There is also some evidence that aerosol can alter the thermodynamics of cloud
formation through semi-direct effects (Sakaeda et al., 2011). Studies using high resolution limited area models have
shown a variety of effects, including stratus to cumulus transition resulting from these interactions (Yamaguchi et al.,
2015; Gordon et al., 2018; Lu et al., 2018). The semi-direct effect has also been shown to be important in a limited
time run of a global model (Das et al., 2020).
During the course of its transport over the Atlantic basin, the dense BB aerosol layer affects the underlying clouds
and Earth's radiative balance in multiple ways. It exerts a direct radiative forcing (DRF) by scattering and absorbing
solar radiation in the atmosphere; when clouds are present, these aerosols absorb incoming solar radiation along with
the radiation reflected by the underlying cloud surface (Chand et al., 2009; Meyer et al., 2013; Zhang et al., 2016).
Simultaneously, depending on the relative vertical location of the aerosol with respect to the cloud deck, the cloud
cover (fraction) or liquid water path may increase or decrease in response to heating of surrounding air masses due to
aerosol absorption and subsequent changes in atmospheric stability, the semi-direct forcing (Sakaeda et al., 2011;
Wilcox, 2012; Das et al., 2020). Observations at Ascension Island show that daytime cloud cover and relative humidity
are lower when there is more smoke in the marine boundary layer (Zhang and Zuidema, 2019). Moreover, as the
marine boundary layer (MBL) deepens farther offshore and north of $5°$ S, subsiding aerosol particles become entrained
into the MBL and interact with the clouds as cloud condensation nuclei to affect their microphysics (indirect forcing)
(Costantino and Breon, 2013; Painemal et al., 2014).
In the context of simulating the above alluded aerosol radiative effects, it is vital that aerosol-cloud overlap
characteristics are accurately represented within the models. The quantification of these aerosol-cloud overlap
characteristics in the models is necessary for a variety of reasons. For example, previous studies have found that the
sign and magnitude of DRF of absorbing aerosol above clouds (AAC) critically depends upon the reflectance and
coverage of the underlying cloud surfaces along with the optical properties, composition and size distribution of the
overlying aerosols (Keil and Haywood, 2003; Chand et al., 2009). Additionally, the magnitude and sign of the aerosol
semi-direct effects are quite sensitive to the vertical distribution of aerosols, especially with respect to the vertical
location of clouds (Penner et al., 2003; McFarquhar and Wang, 2006; Koch and Del Genio, 2010).
Here we address the evolution of the vertical properties of BB aerosol as it travels in the marine environment after
leaving the African land mass. Section 2 identifies the field campaign and specifies the geographic region selected for
the analysis and rationale for that choice. Section 3 describes the attributes of the instrument and key parameters
related to the aerosol that can be extracted from the measurements. Section 4 presents the results followed by a
summary and conclusion in section 5.
**2 Field Campaigns**
The concerns mentioned above were the driving force behind plans for several international multi-year field
campaigns; ORACLES (Observations of Aerosols above Clouds and their intEractionS, Redemann et al., 2021),
CLARIFY-2017 (CLoud-Aerosol-Radiation Interactions and Forcing for Year 2017, Haywood et al., 2021), and
LASIC (Layered Atlantic Smoke Interactions with Clouds, Zuidema et al., 2016, 2018). A key component of the
September 2016 NASA ORACLES Intensive Observation Period (IOP) was the vertical profiling of aerosol properties
measured by an airborne lidar, the NASA Langley High Spectral Resolution Lidar-2, HSRL-2 (Burton et al., 2018),
on-board the NASA ER-2, which was based in Walvis Bay, Namibia, for operations during 2016, the deployment
covered in this study. In the following two years, the instrument was on-board the P-3 flying out of São Tomé. The
siting and flight tracks chosen ensured adequate coverage of the seasonal BB aerosol.
**2.1 Meteorology**
The September monthly mean meteorological situation is shown in Fig. 1 from MERRA2 reanalysis (Buchard et
al., 2017; Randles et al., 2017) along with locations of relevant sites. A thorough meteorological analysis for all
ORACLES deployments is provided in Ryoo et al. (2021). For the period under consideration here, they found that
the African Easterly Jet-South (AEJ-S), fast moving zonal easterlies centered on 650 hPa around 5-15°S, was active
and corresponded closely to the long-term climatology. Fig. 2 shows 650 hPa winds from MERRA2 reanalysis at the
beginning, at the end, and on two intermediate days during which HSRL-2 measurements were made. ER-2 flight
tracks during the September 2016 IOP are shown in Fig. 3. Note that flights were primarily confined to within roughly
1000 km of the African coast with only the 22 September flight venturing further. Flights such as executed during the
IOP are unable to follow air parcels in a Lagrangian fashion to examine the evolution of smoke plumes. Here we
provide an alternate framework by which to study evolving aerosol properties in an average sense. In order to establish
average characteristics of the BB smoke plume as it travels over the ocean, we have chosen five grid boxes of two-
degree latitude and longitude on a side at various distances from the source and aggregated observations. The choice
of grid boxes was based on the availability of data from the flights (Fig. 3) and the general direction of transport of
the smoke as evidenced by the wind fields in Fig. 2. The grid boxes so chosen are marked on Figs. 2 and 3, and the
rationale for the choice is explained below.
Figure 4 shows 48-hour backward trajectory frequency analyses at 3.5 km, roughly the central altitude of the plume,
using NOAA HYSPLIT trajectory calculations (https://www.ready.noaa.gov/HYSPLIT_traj.php) which were carried
out using archived GDAS 0.5-degree meteorology (Stein et al., 2015). The frequency distribution is a 48-hour history
of the paths taken by air parcels arriving at the grid boxes marked A and C at 3500 m altitude. The time period of the
frequency analyses covers the entire period during which HSRL-2 measurements were made, 12-24 September 2016.
The selected grid box pairs indicate that Box A receives aerosol that has earlier crossed Box B and Box C is downwind
of Box D; boxes B and D receive aerosol directly from BB sources on land. The grid box pairs A/B and C/D can
therefore provide information on the evolution of the microphysics and vertical distribution of BB aerosol plumes
after leaving the continent. This strategy is similar to that used in comparisons of models with observations for this
campaign by Shinozuka et al. (2020), who also showed that observations made on the sampled days were
representative of monthly means. In addition to the four boxes strongly influenced by smoke, a southern box, E, has
been chosen to provide a control contrast to the other areas in that it is influenced primarily by maritime air as seen
from Figs. 1 and 2.
**2.2 ORACLES 2016 IOP**
The days during the campaign that were included in the averaging procedure are shown in Table 1. Also included
is the typical time of the day when the measurements were made, a function of the flight pattern of the ER-2. The
number of lidar return profiles averaged for each grid box and statistics related to the backward trajectories are also
listed. These grid boxes contained aircraft tracks on multiple days during which trajectory analysis showed near-
uniform wind direction between 2.5 and 4.5 km altitude throughout the IOP. With the exception of the grid box
centered at $22^\circ$ S, $9^\circ$ E, all indicate flow from the source region of BB aerosol. Table 1 also lists the mean and standard
deviation of time duration in hours spent over water of air parcels arriving at 3500 m altitude at the grid box during
the averaging period. There is no entry for Box E since arriving air had a maritime source and did not originate from
land. It must be stressed that the duration is not calculated from the source region on land, which is distributed over a
large area of central Africa (e.g., Fig. 9 of Redemann et al., 2021) and cannot be uniquely identified with specific
observations made over the ocean. The plume has already been airborne over land for several hours (see Fig. 4) and
aerosol would have undergone transformations that occur at short time scales (Cappa et al., 2020). The duration was
calculated by running HYSPLIT backward trajectories of air parcels arriving every six hours starting at 0600 UTC on
the days of the first flight and ending at 1800 UTC on the days of the last flight of the averaging period and is shown
in some detail in Fig. 5, which essentially reflects the profile of the prevailing wind speeds. The inference is that BB
smoke at 3500 m altitude arrives at A on average about 30 h after passing B and arrives at C 35 h after passing D. The
change in selected aerosol properties as measured by the HSRL-2 during this travel in the marine environment provides
information on the evolution of the plume during this time period.
**3 HSRL-2**
The NASA LaRC HSRL-2 uses the HSRL technique to independently retrieve aerosol extinction and backscatter
(Shipley et al., 1983; Grund and Eloranta, 1991; She et al., 1992) without a priori assumptions on aerosol type or
extinction-to-backscatter ratio. By using the HSRL technique, HSRL-2, like its predecessor HSRL-1 (Hair et al.,
2008), provides accurate backscatter profiles even in situations where the lidar beam is attenuated by overlying cloud
or aerosol as long as it is not completely attenuated. The LaRC HSRL-2 employs the HSRL technique at 355 and 532

nm, and the standard backscatter technique at 1064 nm. It also measures aerosol and cloud depolarization at all three wavelengths. The HSRL-2 provides vertically resolved measurements of the following extensive and intensive aerosol parameters below the aircraft (approximate archival horizontal, $\Delta x$, and vertical resolutions, $\Delta z$, are listed assuming ER-2 cruise speed).

• *Extensive parameters*[1] – backscatter coefficient, $\beta$, at 355, 532, and 1064 nm ($\Delta x \sim 2$ km, $\Delta z \sim 15$ m); extinction coefficient, $\alpha$, at 355, and 532 nm ($\Delta x \sim 12$ km, $\Delta z \sim 300$ m); optical depth at 355 and 532 nm (integrating the profile of extinction). The aerosol optical depth (AOD) is a critical quantity in discussions of the influence of aerosol on climate (Boucher et al., 2013).

• *Intensive parameters* – extinction-to-backscatter ratio of aerosol, the Lidar Ratio, $S_a = \alpha_a / \beta_a$, at 355 and 532 nm ($\Delta x \sim 12$ *km*, $\Delta z \sim 300$ m); depolarization, $\delta_a = \beta_a^\perp / \beta_a^\parallel$, at 355, 532, and 1064 nm ($\Delta x \sim 2$ km, $\Delta z \sim 15$ m); and aerosol backscatter wavelength dependence (i.e., Ångström exponent for aerosol backscatter – directly related to the backscatter color ratio) for two wavelength pairs (355-532 and 532-1064 nm, $\Delta x \sim 2$ km, $\Delta z \sim 15$ m).

The overall systematic error associated with the backscatter calibration is estimated to be less than 5 % for the 355 and 532 nm channels and 20 % for 1064 nm (Burton et al., 2015). Under typical conditions, the total systematic error for extinction is estimated to be less than 0.01 km$^{-1}$ at 532 nm. The random errors for all aerosol products are typically less than 10 % for the backscatter and depolarization ratios (Hair et al., 2008). Rogers et al. (2009) validated the HSRL extinction coefficient profiles and found that the HSRL extinction profiles are within the typical state-of-the-art systematic error at visible wavelengths (Schmid et al., 2006). Since HSRL-2 includes the capability to measure backscatter at three wavelengths and extinction at two wavelengths, "3β+2α" microphysical retrieval algorithms (Müller et al., 1999a, 1999b; Veselovskii et al., 2002) are used to retrieve height-resolved parameters such as aerosol effective radius and number, surface, and volume concentrations (Müller et al., 2014, Sawamura et al., 2016). Here we restrict ourselves to the effective radius of the particles.

**4 Results**

In this study of the vertically resolved evolving properties of BB aerosol, we present key lidar measurements and microphysical results obtained by performing the "3β+2α" retrieval mentioned in Section 3.

---

[1] By the term *extensive*, we refer to optical parameters, such as extinction, that are influenced by the amount (concentration) and type (size, composition, shape) of aerosol/cloud particles. *Intensive* properties, on the other hand, are those that depend only on the nature of the particles and not on their quantity or concentration, but rather depend only on aerosol type (Anderson et al., 2003).

**4.1 Lidar**

Vertical profiles averaged over the times of overflight in 2°×2° latitude/longitude boxes shown in Figure 3 on the days given in Table 1 are for the following properties.

1. Aerosol Extinction at 532 nm, $\alpha_a$, determined by aerosol number concentration, microphysical properties and relative humidity

2. Backscatter Ångström exponent between 1064 and 532 nm, an indication of particle size.

3. Aerosol Depolarization at 532 nm, a measure of particle asphericity.

4. Aerosol extinction to backscatter ratio, the Lidar Ratio, at 532 nm, a marker for aerosol composition.

Inspection of the wind field at 650 hPa in Fig. 2 and backward trajectory frequency plots in Fig. 4 suggest that the grid boxes chosen fit naturally into two pairs of tracks of the widespread BB aerosol field. The northern pair, identified in Table 1 as A and B, centered around 10° S, are in a faster zonal track, whereas the grid boxes C and D are in a track centered between 13-15° S that is slightly slower and has a component from the north over a stretch of water (Fig. 2). The two pairs can then provide information on the evolution of aerosol properties over a time scale of one to two days. Figures 6-9 show the aerosol extinction, backscatter Ångström exponent, aerosol depolarization and Lidar Ratio for the two pairs of grid boxes and Box E, which is at the southern edge of the region influenced by the BB aerosol. The results presented are one-minute averages of independent 10 s vertical profiles for backscatter Ångström exponent and depolarization and one-minute averages for extinction and lidar ratio profiles. From Table 1, the mean time elapsed between B and A is 29.4 h and that between D and C is 34.9 h. It should be pointed out that parameter values shown below the level of mean cloud top are averages of lidar returns through breaks in the stratus deck and are not relevant for this study. If we use the low cut-off of an extinction coefficient of 15 Mm$^{-1}$ to indicate an aerosol-free layer (Shinozuka et al., 2020), then Fig. 6 indicates that the bulk of the smoke layers encountered at these distances from land were separated from the cloud top, a feature more prevalent during the 2016 IOP than in 2017 and 2018 (Redemann et al., 2021).

The northern plume is a column of aerosol of relatively constant extinction from just above 2.5 km to 5 km while the southern plume has a profile of extinction that increases nearly linearly with height from a minimum near the cloud top to a maximum at 5 km (Fig. 6). The vertical structure of the aerosol profiles measured by HSRL-2 was compared to water vapor profiles represented by the Modern-Era Retrospective analysis for Research and Applications, Version 2 (MERRA2) model. Pistone et al. (2021) explored the relationship between aerosols, CO, water vapor as measured by ORACLES airborne in situ measurements and represented by models including MERRA2. They found the MERRA2 water vapor profiles, like the measured water vapor profiles, exhibited a linear relationship with CO and biomass burning plume strength; they also found that smoky, humid air produced by daytime convection over the continent advected over the ocean and into the ORACLES study region. MERRA2 water vapor profiles produced at three hourly increments and 72 pressure levels were interpolated to the times and locations of the HSRL-2 profiles. Water vapor mixing ratio generally decreased significantly just above the PBL then increased for altitudes around 2 to 3 km before decreasing again. This behavior is generally consistent with the relationship between water vapor and aerosol scattering reported by Pistone et al. (2021).

Figure 10 shows the median, 25th and 75th percentile relative humidity (RH) profiles computed by interpolating the
MERRA2 0.5-deg. 3-hourly humidity profiles to the locations and times of the HSRL-2 measurements. The profiles
typically show a more pronounced increase in RH with altitude that more closely follows the HSRL-2 measurements
of aerosol extinction profiles, although the MERRA2 profiles typically begin decreasing above 4 km whereas the
airborne in situ RH measurements and HSRL-2 aerosol extinction profiles begin decreasing above 5 km. Interestingly,
during three of the dates (Sept. 12, 16, 22) considerable portions of the smoke layers correspond to MERRA2 relative
humidity above 60-70%. This increase in RH with altitude could help explain at least some of the increase in aerosol
extinction with height observed in the HSRL-2 profiles of the C/D Box pair. Aerosol humidification often amplified
the increase in aerosol extinction by factors of 1.5 or more (Doherty et al., 2022).
The Ångström exponent (Fig. 7) and depolarization (Fig. 8) indicate the presence of fine spherical particles at the
top of the plume and increasing sizes towards the bottom. The Lidar Ratio (Fig. 9) above 3 km for the two pairs is
between 70 and 80 sr, suggesting strong absorption (Müller et al., 2019) but is considerably less and highly variable
in Box E and in the lower layers of the aerosol plume in Box D, where the smoke plume most likely has components
of continental aerosol such as dust and pollution typical of the nearby Namibian coast (Klopper et al., 2020). The most
striking feature of the results is the very small profile-to-profile variability of the intensive lidar parameters in the
upper two kilometers of the plume over the course of several days as evident from the range of values in the 25-75
percentile shaded grey in Figs. 7-9. This suggests strongly that the particles maintain their size, shape and absorbing
properties over the first few days of transport over the ocean. This result is of some importance for climate studies in
which the radiative properties of BB aerosol are input to the calculation of radiative forcing. Complex chain aggregates
as found near the source of fires (Pósfai et al., 2003, China et al., 2013) are typically not represented in climate models.
However, if the aerosol is already spherical and maintains its size over the time period of radiative interactions being
studied, then core-shell models of varying degrees of complexity could perhaps suffice (Zhang et al., 2020). The lower
portion of the plume containing larger BB aerosol particles is subject to mixing with marine and continental particles
from regions not affected by biomass burning and is highly variable in nature. This would be more difficult to model
but Fig. 6 shows that the aerosol extinction coefficient decreases rapidly at lower levels so errors in representation
may be acceptable.
**4.2 Microphysics**
The lidar measurements are inverted to obtain information regarding particle size. The inversion is performed on one-
minute averages of six independent 10 s backscatter profiles and one-minute average extinction profiles. Details of
the inversion process are in Müller et al. (2019) and references therein. The particle size distribution is represented
using a series of eight triangular basis functions that can represent both monomodal and bimodal size distributions
(*ibid*). Points to note are that the procedure makes the following assumptions: the particles are spherical and
homogeneous having wavelength-independent complex index of refraction. The low (< 5 %) values of depolarization
through most of the plume, shown in Fig. 8, suggests that the spherical assumption is justified. There is most likely
structure and inhomogeneity in the core of the particles, but current particle optical models are unable to incorporate
these complexities. Results from this inversion procedure have been compared to coincident airborne in situ particle
measurements. Müller et al. (2014) present results from a campaign off the northeast coast of the US showing that the
inversion results agree with in situ measurements of effective radius and also number, surface area and volume
concentration within error bars. Sawamura et al. (2017) report on campaigns in the wintertime San Joaquin Valley of
California and summertime near Houston, TX. They found high correlation and low bias in surface and volume
concentration in situ measurements relative to HSRL with the best agreement for submicron fine-mode aerosol, which
is most relevant to the current study. Müller et al. (2019) report retrievals and their uncertainty for one day in the
ORACLES campaign, 22 September 2016. Considering only optical data with strong signal-to-noise ratio, they
estimate retrieval errors are 25 % for number concentration. The relative uncertainty in effective radius for parts of
the flight track where particle size was nearly constant was below 20 %.
In order to help separate particles that have BB source from coarser particles of continental or marine origin, we
specify a Submicron Fraction (SMF) as the contribution to the extinction at 532 nm of particles in the radius range
0.05-0.50 μm (Anderson et al., 2005). Figure 11 shows the profiles of SMF for the five grid boxes and not surprisingly,
the bulk of the smoke plume, especially between 3 and 5 km contains aerosol almost entirely in the submicron range.
Below 3 km, at locations both near and further way from the coast, there is a marked increase in the fraction of larger
particles. The increase in depolarization (Fig. 8) at these lower levels and a decrease in the Lidar Ratio (Fig. 9) suggest
mixing with the aforementioned non-BB aerosol particles. However, the sharp decrease in extinction below 3 km (Fig.
6) indicates that their contribution to direct radiative effects would be minimal. Finally, Fig. 12 shows the vertical
profile of the effective radius of the SMF aerosol population. The effective radius is 0.16 μm with little variation
between 3 and 5 km. Of greater significance is that it remains very similar between the pairs of grid boxes along the
transport trajectory of the smoke. The retrieved effective radius is similar to the results presented by Müller et al.
(2014) for a mixture of urban aerosol and smoke. Their comparison with in situ measurements showed a slight
overestimate but within a standard deviation. The retrieved and in situ results also show that the particle size is uniform
with altitude even when the number concentration drops by a factor of three. Another set of prior comparisons of
HSRL-2 and in situ measurements is provided in Sawamura et al. (2017). Here again, the effective radius of the
submicron fraction of particles, 0.15 μm, is uniform with altitude, and comparable though biased slightly low
compared to in situ observations.
The effective radii of the SMF aerosols, which typically vary between 0.15 to 0.20 μm, are generally consistent
with the sizes reported previously for smoke aerosol in the ORACLES region. Haywood et al. (2021) provide a
composite of the aerosol sizes for biomass burning aerosol off the South African coast. These size distributions, which
were derived from airborne in situ measurements (Haywood et al., 2003; Peers et al., 2019; Wu et al., 2020), typically
correspond to SMF aerosol effective radii between 0.14-0.17 μm and were for the dry aerosol. Shinozuka et al. (2020)
reported on airborne aerosol sizes measured during ORACLES by an Ultra-High-Sensitivity-Aerosol Spectrometer
(UHSAS) deployed on the NASA P-3 aircraft. The UHSAS measured particles with dry diameters between 60 and
1000 nm. SMF aerosol effective radii derived from the UHSAS measurements of volume mean diameter were
generally around 0.09-0.10 μm for the dry aerosol. Shinozuka et al. (2020) noted that the UHSAS measurements were
somewhat undersized and so were adjusted to account for this effect; this adjustment improved scattering closure with
coincident nephelometer measurements. As discussed in section 4.1, the RH on some days was above 60-70% so that
effective radii under ambient conditions could be expected to be somewhat higher than for the dry aerosol. Using
measurements from an airborne Differential Aerosol Sizing and Hygroscopicity Spectrometer Probe (DASH-SP),
Shingler et al. (2016) quantified the size-resolved growth factors for several aerosol types; they found that at RH~70-
80%, particle diameters for biomass burning aerosols were about 15-20% larger than for the dry aerosol. Xu et al.
(2021) derived aerosol properties during the 2016 ORACLES mission using an inversion algorithm that combined
HSRL-2 and Research Scanning Polarimeter (RSP) remote sensing measurements. These retrieved aerosol properties
were then compared with those derived from the UHSAS measurements described by Shinozuka et al. (2020). For
measurements acquired on Sept. 12, 2016, the SMF aerosol effective radius derived from the remote sensing
measurements were generally between 0.12-0.15 μm and were only slightly (0.012 μm) higher than the effective radii
for the (dry) SMF aerosol derived from the UHSAS measurements. This suggests that some of this difference is
associated with differences in RH between the remote sensing retrievals and the in situ measurements.
**5 Conclusions**
The results of the aggregated HSRL-2 profiles during the 2016 ORACLES IOP presented here show two main
findings. These are however limited to a brief period in the transport of BB smoke from continental Africa over marine
clouds in the Atlantic Ocean. This is a limitation of the 2016 campaign because the flight tracks remained within 1000
km of the coast. For the period of one to two days after crossing the land-ocean boundary, the fraction of all particles
that are in the submicron range in the main smoke plume between 3 and 5 km is around 95 %. The effective radius of
the submicron particles in this altitude interval is 0.16 μm and essentially constant with altitude. The particle size is
comparable to measured particle sizes in previous campaigns that sampled aerosol that was a mixture of urban haze
and smoke (Müller et al., 2014; Sawamura et al., 2017). Moreover, the shape of the median vertical profile of
extinction does not change during the first two days of transport over water suggesting the absence of dry deposition
and wet scavenging. The low (< 0.05) depolarization ratio of the submicron particles signifies that they are well coated
and the assumption of sphericity in the inversion procedure and models that estimate the radiative effects of aerosol
is justified. The BB aerosol mix with continental and marine aerosol at the base of the plume but during the September
2016 IOP this layer of mixed aerosol tended to have very low extinction coefficients suggesting low abundance and
there was often a distinct gap between the plume and the cloud tops.
The HSRL-2 instrument was also deployed in the 2017 and 2018 ORACLES campaigns but was deployed on the
NASA P-3 which often flew at low altitude to acquire in situ measurements of aerosols and clouds. Consequently, the
HSRL-2 was not able to make continuous measurements of the BB aerosol plumes in a manner similar as when
deployed on the ER-2. However, there are segments of the track that can provide similar information to the data
obtained in the 2016 campaign but for a different time period. Moreover, some flight tracks extended much further
from land (Doherty et al., 2021). Analysis of the later campaigns will provide information on the physical evolution
of aerosol that has aged for a longer period than is covered in this study.

**Data Management**

HSRL-2 optical data and retrieved inversion data are available at the NASA archive site https://espoarchive.nasa.gov/archive/browse/oracles/id8/ER2 and are permanently archived at doi:10.5067/SUBORBITAL/ORACLES/ER2/2016_V1.

**Acknowledgements**

The lead author would like to thank NASA Langley Research Center for hosting him during a sabbatical when this study was initiated. HSRL-2 participation in ORACLES was supported by NASA through the Earth Venture Suborbital-2 (EVS-2) program (grant no. 13-EVS2-13-0028). Funding for this work was also provided by NASA through the Radiation Sciences Program. We wish to thank the NASA ER-2 pilots and ground crew for their extensive support during ORACLES.

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

**Table 1:** Averaging area, flight time periods, the duration over water and number of HYSPLIT backward trajectories, and number of HSRL-2 profiles in each grid box used in the study.

| Box | Averaging Area | Averaging Days | Time of Day | Duration in Hours Over Water at 3.5 km | Number of Profiles |
|---|---|---|---|---|---|
| A | 11° S-9° S; 1° W-1° E | 9/12,16 | 11:00 UTC | 44.3±7.0 (N = 19) | 50 |
| B | 10° S-8° S; 8° E-10° E | 9/12,16,18 | 10:00 UTC | 14.9±4.5 (N = 27) | 56 |
| C | 16° S-14° S; 4° E-6° E | 9/12,16 | 13:00 UTC | 40.4±7.2 (N = 19) | 51 |
| D | 14° S-12° S; 10° E-12° E | 9/18,24 | 09:00 UTC | 5.5±2.0 (N = 27) | 46 |
| E | 23° S-21° S; 8° E-10° E | 9/20,22 | 14:00 UTC | - | 36 |



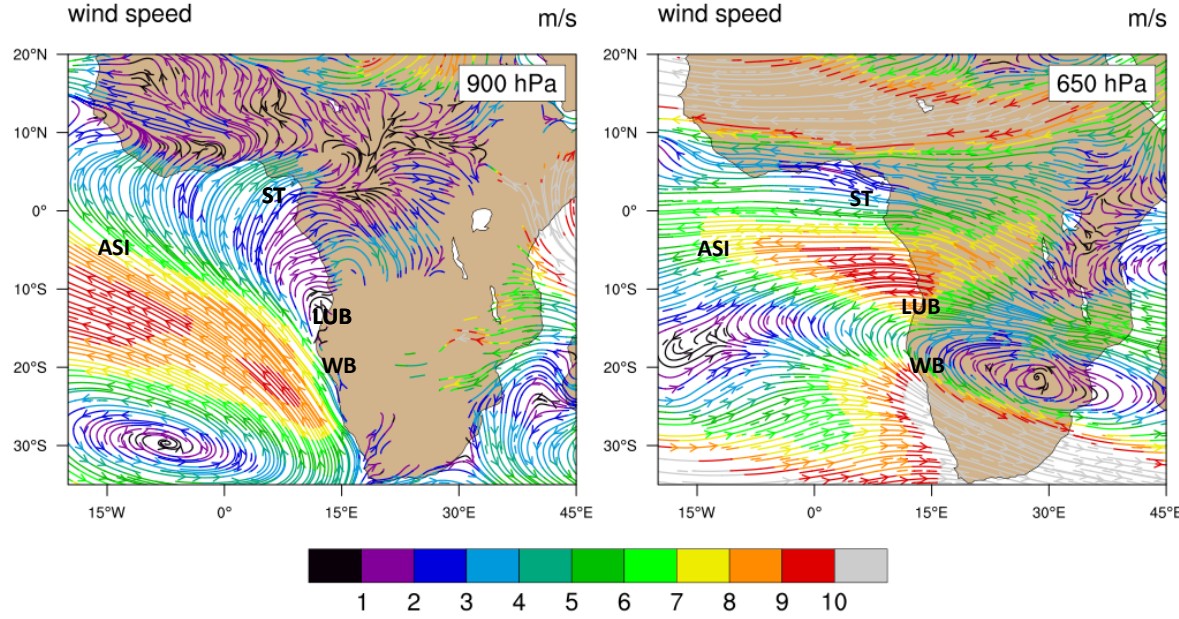

**Figure 1:** MERRA2 monthly mean reanalysis of 900 and 650 hPa streamlines for September 2016. Stations marked
are Ascension Island (ASI), Lubango (LUB), a long-term AERONET site at 2 km elevation, and Walvis Bay (WB),
where ER-2 flights originated from during the September 2016 ORACLES IOP. Flights in August 2017 and
September/October 2018 originated from São Tomé (ST).

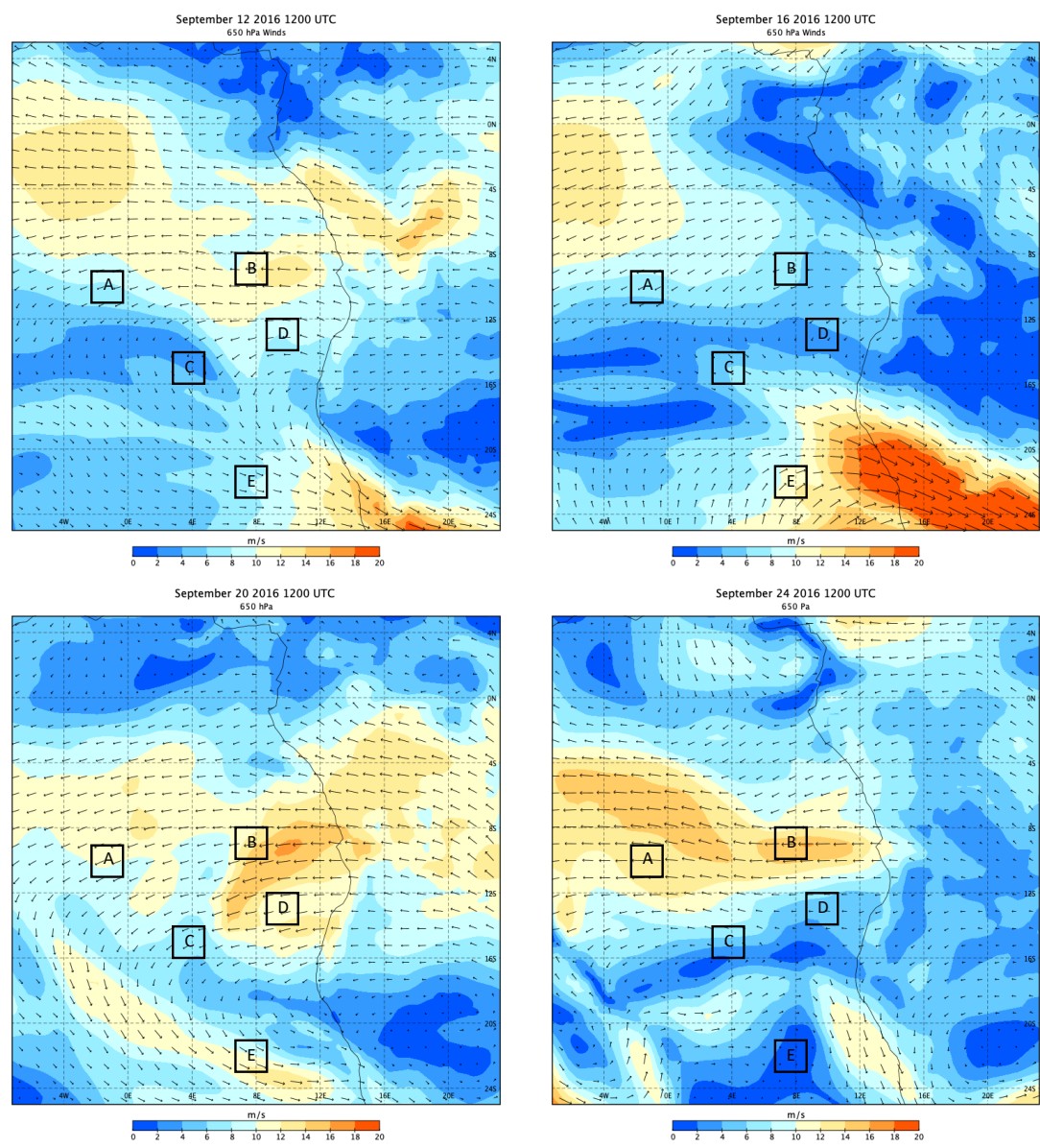

**Figure 2:** MERRA2 reanalysis of 650 hPa winds at 1200 UTC on September 12, 16, 20, 24, 2016. Grid boxes in the study are marked with letters.


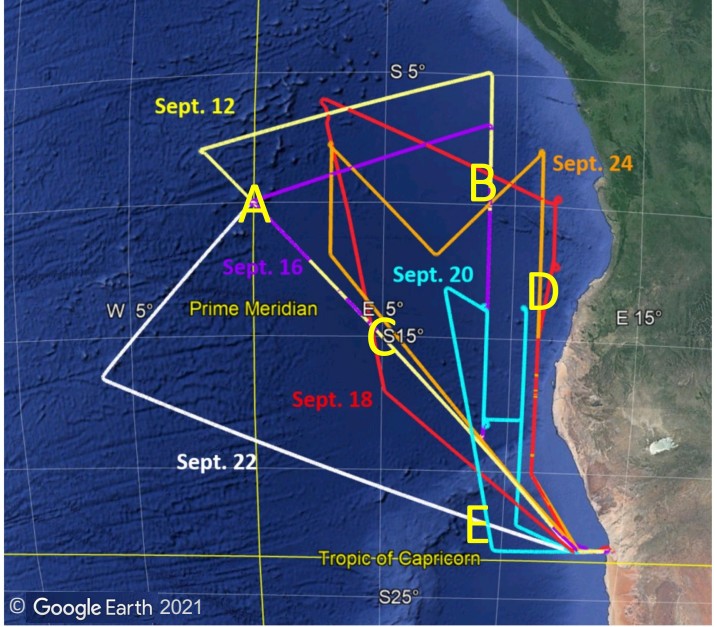


**Figure 3:** HSRL-2 science data flight tracks during the September 2016 IOP. Letters refer. to the grid
boxes identified in Fig. 2 (© Google Earth).

NOAA HYSPLIT MODEL - TRAJECTORY FREQUENCIES
# trajs passing through grid sq./# trajectories (%)    0 m and 99999 m
Integrated from 1200 24 Sep to 1800 10 Sep 16 (UTC) [backward]
Freq Calculation started at 0000 00     00 (UTC)

NOAA HYSPLIT MODEL - TRAJECTORY FREQUENCIES
# trajs passing through grid sq./# trajectories (%)    0 m and 99999 m
Integrated from 1200 24 Sep to 1800 10 Sep 16 (UTC) [backward]
Freq Calculation started at 0000 00     00 (UTC)

**Figure 4:** Frequency distribution of 48-hour backward trajectories of air parcels arriving at 3500 m above the centers of grid boxes A and C over the time period of the campaign. Grid boxes B and D are upstream of grid boxes A and C, respectively.

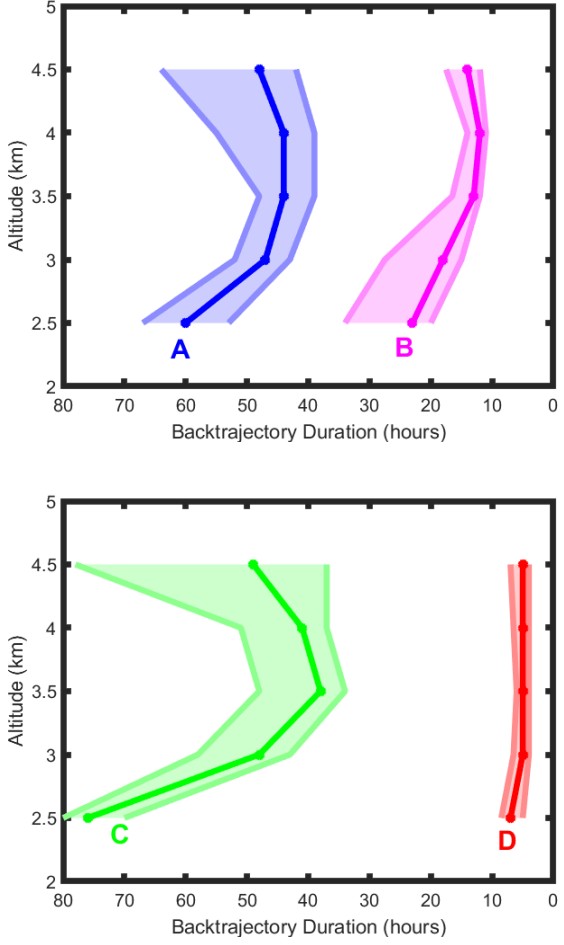

**Figure 5:** Duration of time spent over water of air parcels arriving at grid boxes marked on the figure. Solid lines are median values, and the shaded portion are the range of the 75[th] and 25[th] percentile. The number of trajectories used for the calculation are in Table 1. Trajectory hours are shown in reverse to correspond to the map in Fig. 4.

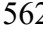




**Figure 6:** Average vertical profiles of the aerosol extinction coefficient at 532 nm in grid boxes A (upper left), B (upper right), C (middle left), D (middle right) and E (lower left). The averaging area, dates of flights and total number of one-minute profiles are also shown. The dark line represents the median value and grey shades contain the 25[th] to 75[th] percentiles. Dashed line refers to the mean cloud top height.


**Backscatter Ang. Expo. 1064/532 Sept. 12,16, 2016**
**lat -11 to -9 lon -1 to 1**

**Backscatter Ang. Expo. 1064/532 Sept. 12,16,18 2016**
**lat -10 to -8 lon 8 to 10**

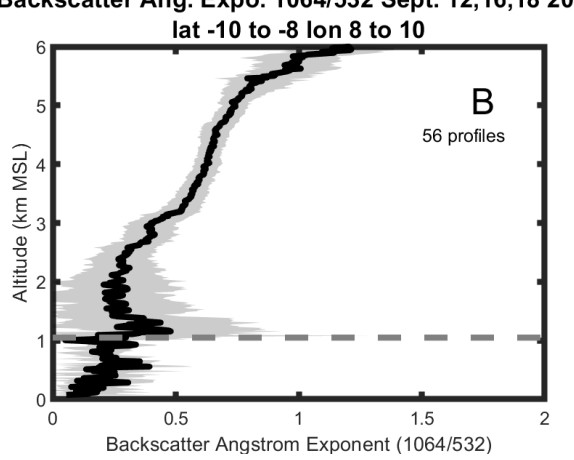


**Backscatter Ang. Expo. 1064/532 Sept. 12,16 2016**
**lat -16 to -14 lon 4 to 6**

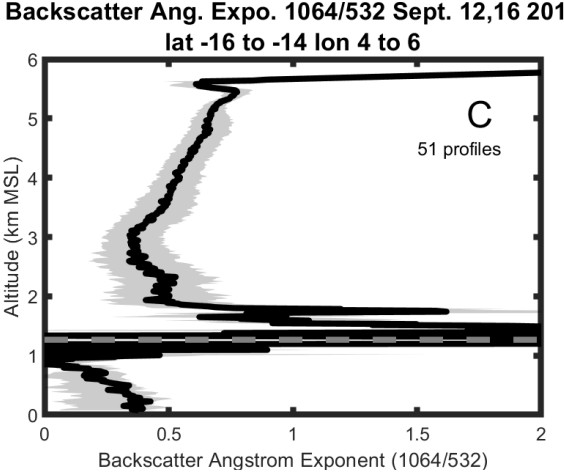

**Backscatter Ang. Expo. 1064/532 Sept. 18,24 2016**
**lat -14 to -12 lon 10 to 12**

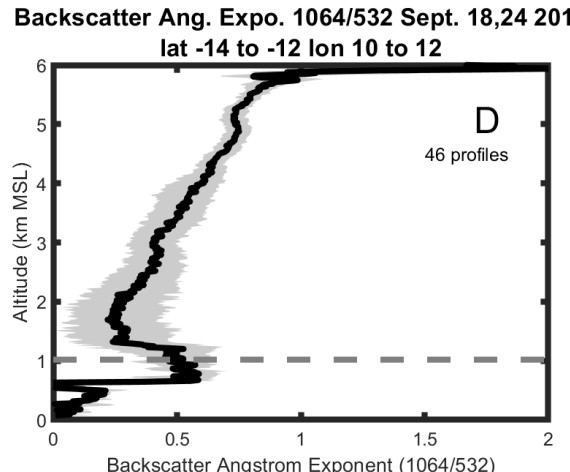


**Backscatter Ang. Expo. 1064/532 Sept. 20,22 2016**
**lat -23 to -21 lon 8 to 10**

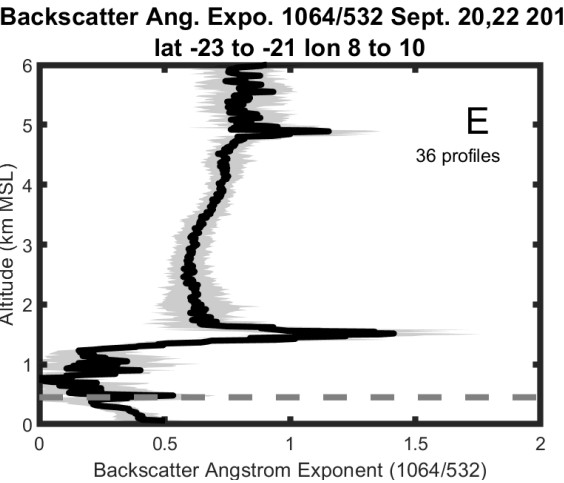

**Figure 7:** As in Fig. 6 but for the Wavelength Dependent Backscatter Ångström exponent between 1064 and 532
nm.


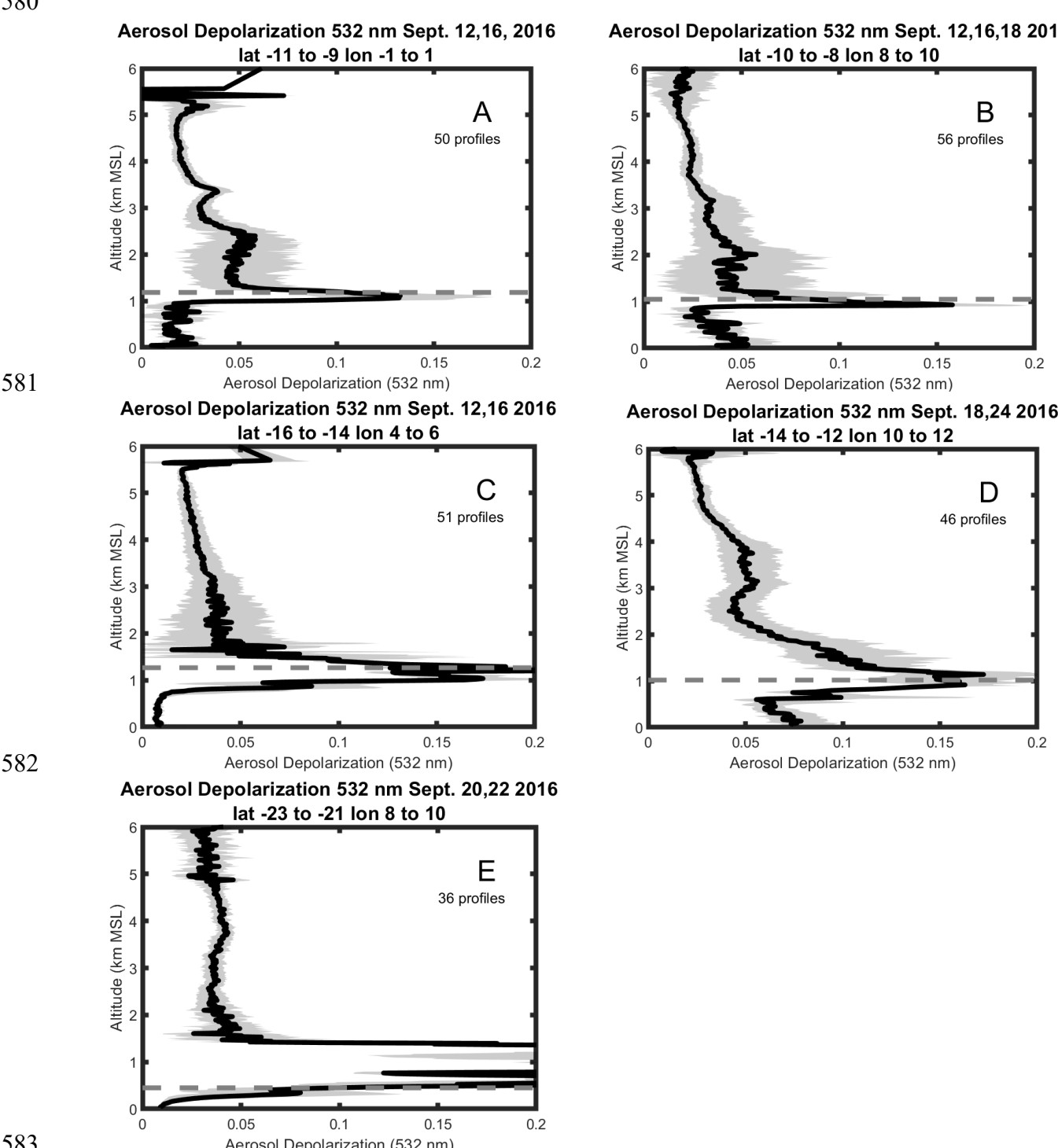

**Figure 8:** As in Fig. 6 but for the aerosol depolarization at 532 nm.


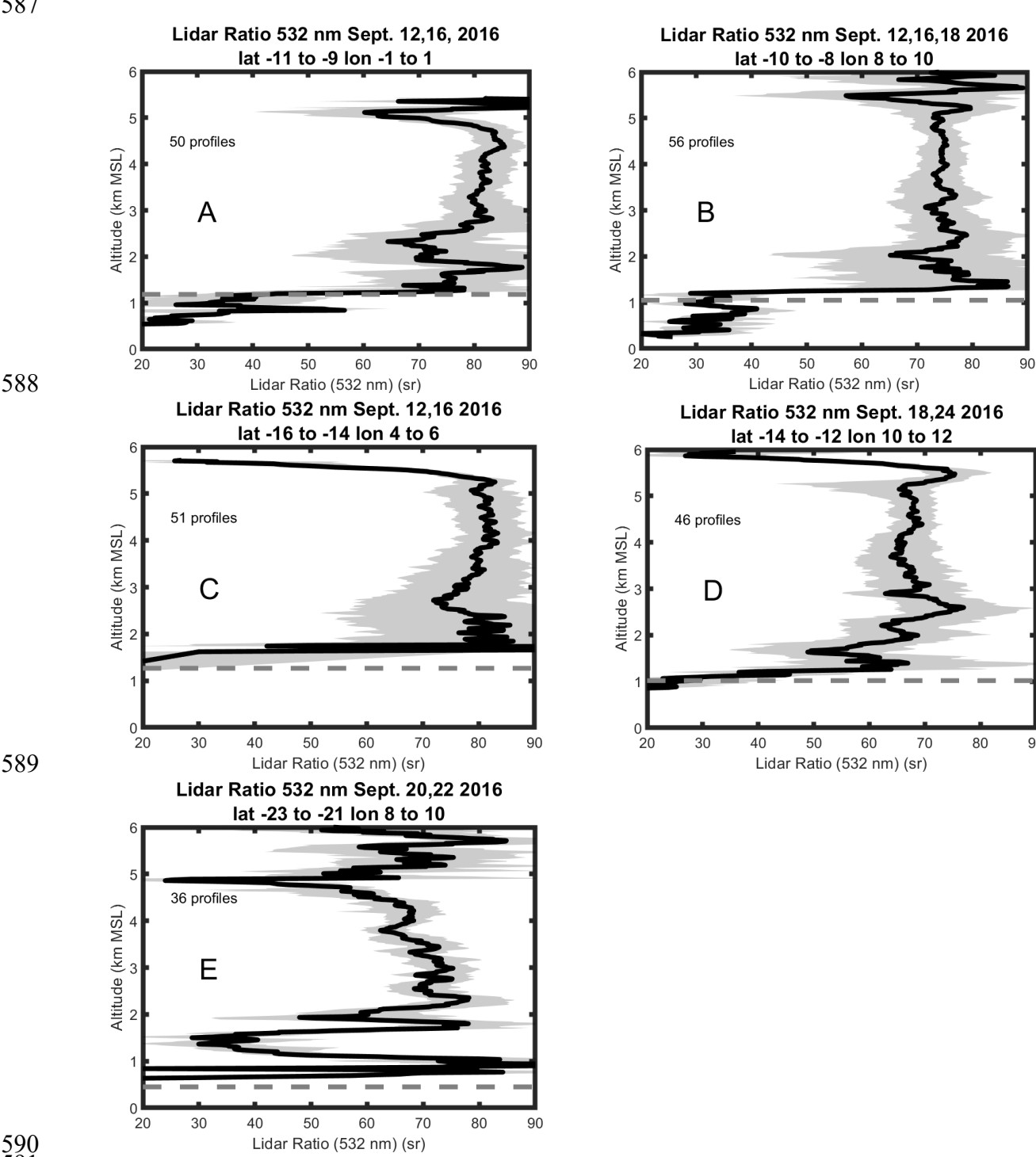



**Figure 9:** As in Fig. 6 but for the Lidar Ratio at 532 nm.

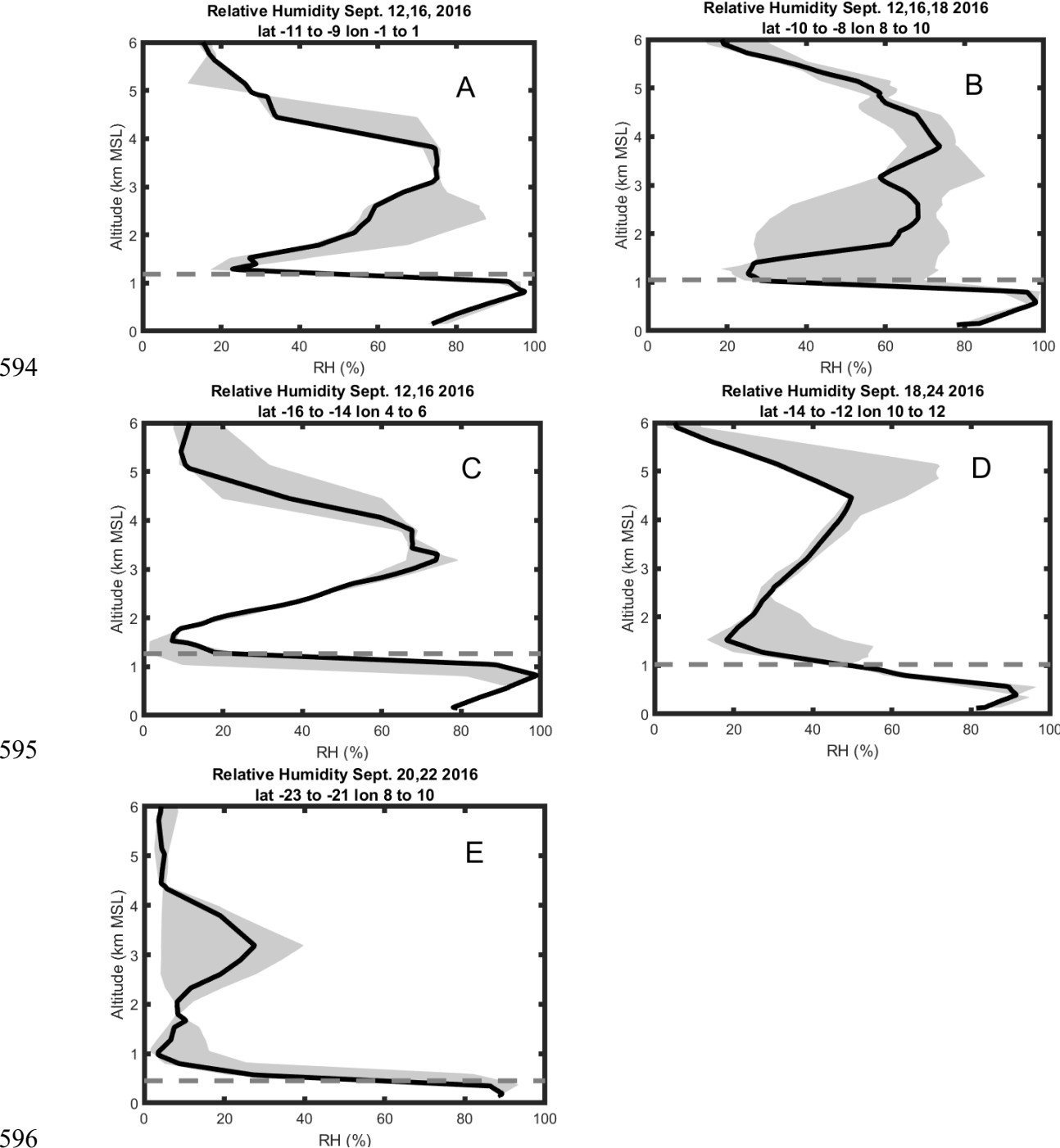




**Figure 10:** Relative Humidity (%) in grid boxes A (upper left), B (upper right), C (middle left), D (middle right) and E (lower left) from MERRA2 reanalysis corresponding to the HSRL-2 profiles shown in Figs. 6-9. The dark line represents the median value and grey shades contain the 25th to 75th percentiles. Dashed line refers to the mean cloud top height.

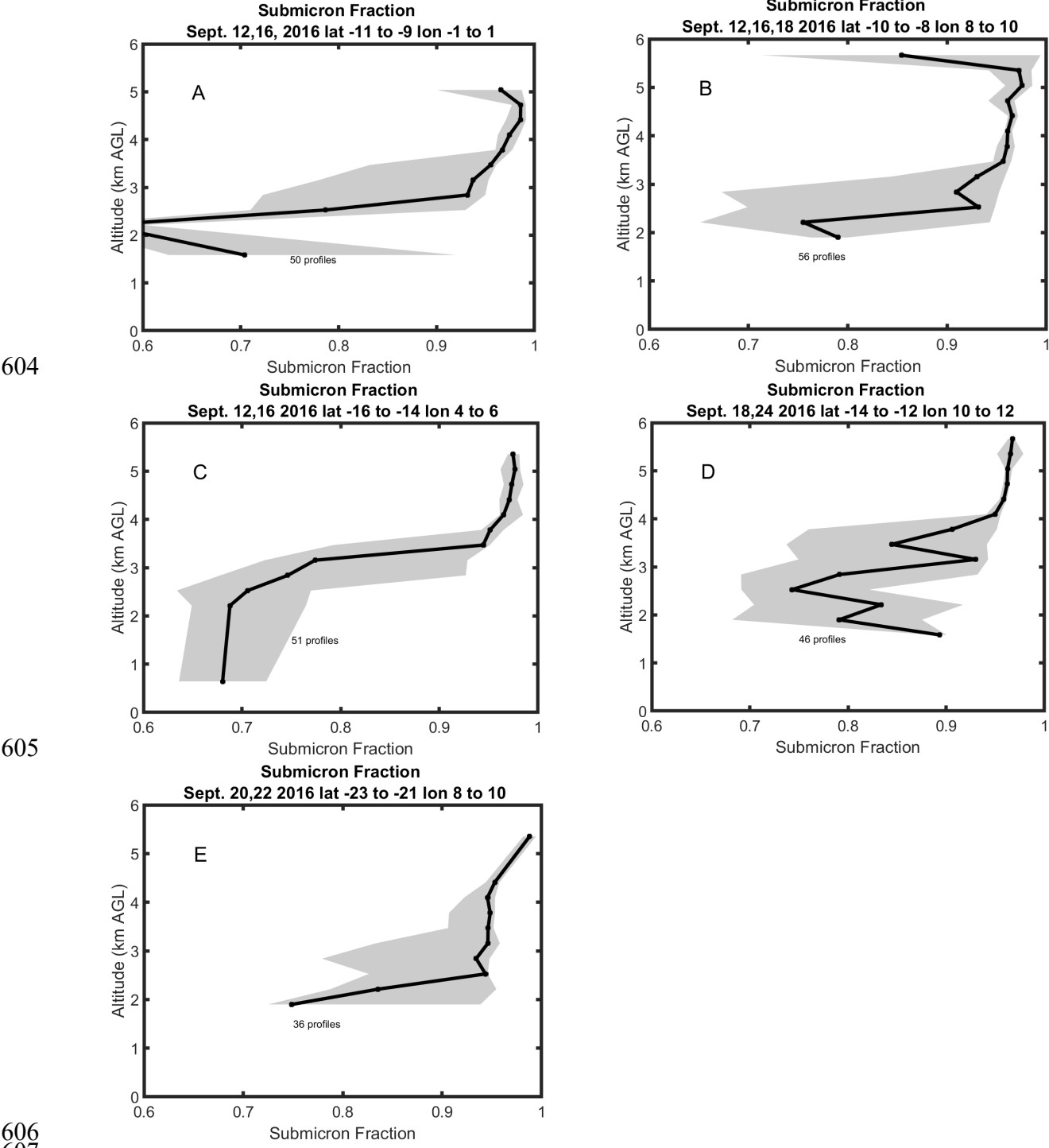




**Figure 11:** Average vertical profiles of the submicron fraction in grid boxes A (upper left), B (upper right), C (middle left), D (middle right) and E (lower left). The averaging area, dates of flights and total number of one-minute profiles in the average are also shown. The dark line represents the median value and grey shades contain the 25th to 75th percentiles.


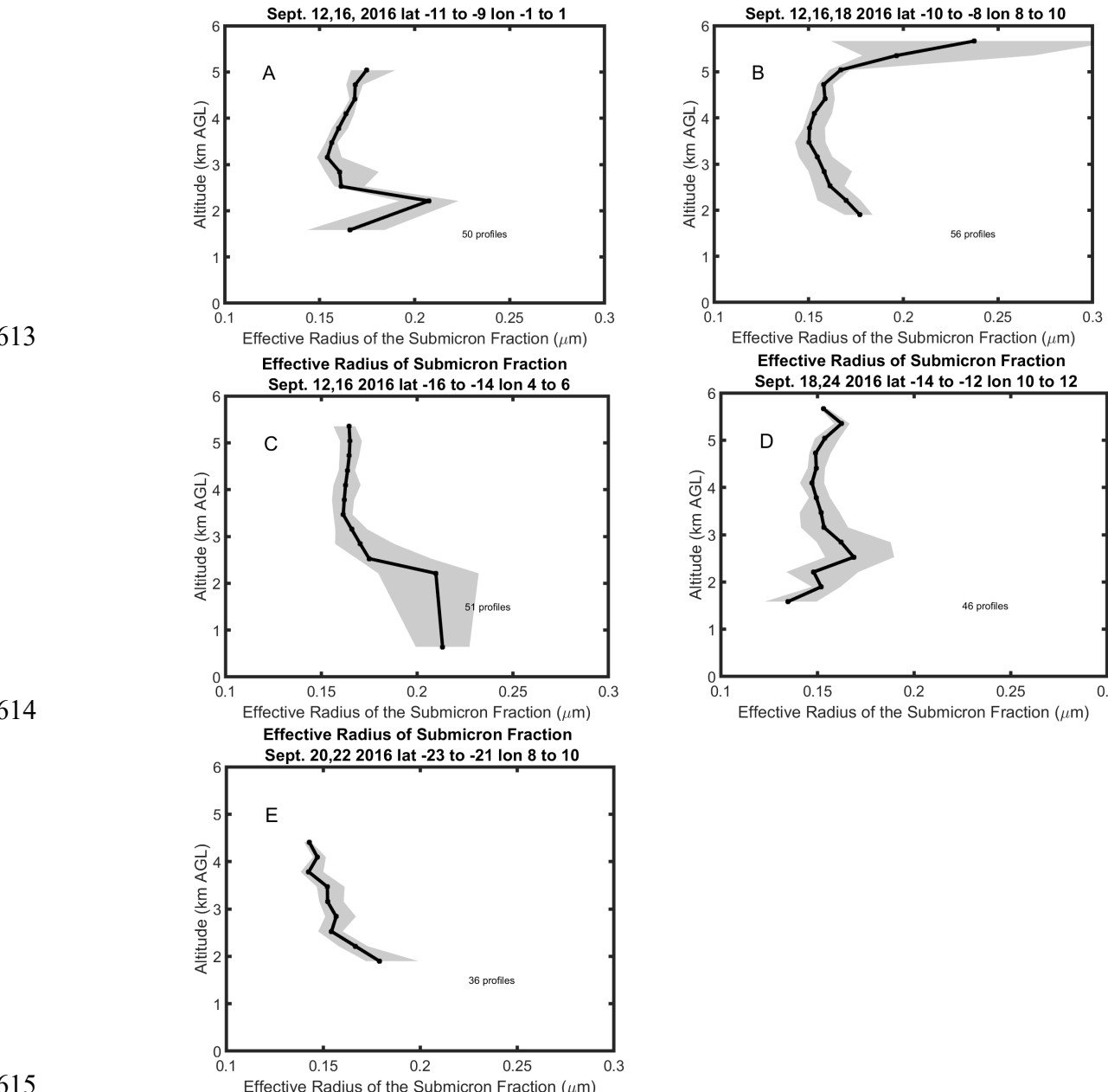



**Figure 12:** As in Fig. 11 but for the effective radius of the submicron fraction.