# Peer review of "Vertical structure of biomass burning aerosol transported over"

_Atmospheric Chemistry and Physics, 2021_

## Referee Comment (RC1)

Review of:
Vertical structure of biomass burning aerosol transported over the southeast Atlantic Ocean
by: Harshvardhan et al.
Manuscript doi:10.5194/acp-2021-846

This study examines HSRL2 profiles over the southeast Atlantic and interprets them for insights into aerosol microphysical behavior and transport. This is a valuable objective given the quality of the data, its relevance for satellite data interpretation, and the dominance of smoke above clouds for this region. The work hasn't been carried through as completely as it needs to be for publication, however. Major requests to the authors include the following:

- we need more synoptic context for the days forming the data sample. When was the African Easterly Jet-south active? Did mid-latitude disturbances ever constrain the aerosol to be closer to the coast?
- How do the Lidar profiles correspond with the humidity structure, in particular the extinction? Does this explain the vertical increase in extinction with height? You can use MERRA2 or ERA5 to get the humidity structure, and the Pistone et al. 2021 paper to rationalize the choice.
- the figure panel titles are often difficult to read and should be improved.

Recommendation: accept with major revisions.

More specific comments are shown below, some of which are repetitive of each other and/or the major comments above.

Abstract:
Line 15: Should HSRL ->HSRL-2 ?
Last sentence: it would be worth repeating here that the 'clear' gap is associated with September.

Introduction:
Line 32-33: It would be nice to find a quote from the most recent IPCC AR6 to replace this one.

Line 41: the recent paper by Kacarab et al., 2020 would serve to underpin this statement. Overall, it would be good to include some updated references. See the ACP special issue.

Lines 56-63: how about also highlighting the ability to discern aerosol-cloud contact? This will affect the reflectance of the underlying cloud surfaces.

Lines 75: the appropriate reference for CLARIFY is Haywood et al., 2021.

Line 85: frame => framework

Lines 92-101: we need more information on why grid boxes A and C were chosen. How much data fell into these boxes and from which days? What distinguishes the synoptics establishing A-B versus C-D? fig. 3 only shows the monthly-mean fields. See Ryoo et al., 2021.

Line 147: I think you just show the effective radius, from this list of possible retrieved variables. Would be good to clarify.

Lines 154-157: yet this list is missing some of the variables you do show….

Line 160: the dates and amount of data from each contributing flight should be provided for each of the boxes. Figure 2 indicates them, but it is qualitative. it might be helpful to show the 2 by 2 degree boxes on fig. 2. Do the northern pairs correspond to when the African easterly jet-south was active? September is also a month when mid-latitude disturbances to the south constrain aerosol to closer to the coast (Ryoo et al., Zhang and Zuidema 2021) and the anomalous ascent can help mix the aerosol (and humidity) upwards - that might explain why the profiles in C-D show the more linear increase in extinction with height. This would also help explain the increase in the depolarization ration below 2km for D .

Line 172-174: interesting, that the northern and southern plumes differ in this fashion. Do you have an idea why? Is the extinction correlated with humidity? You could use the MERRA2 relative humidity profiles to ascertain.

Fig. 2: explain in caption what the letters within the figure mean. Label latitude lines

Figure 3: use the same spatial domain for the top and bottom figures. What do hysplit back trajectories at 2.5 km look like? Do those also connect to land? I ask because the back trajectory duration is so much longer for those in fig 4. This would also help explain if it is marine or continental air with different aerosol properties getting incorporated into the retrievals. Would a case study be helpful?

Fig. 5: It would be helpful to put these time ranges in synoptic context. Visit Ryoo et al., 2021.

What was the relative humidity and how does it affect these results?

Fig. 7: it should be possible to discriminate whether the enhanced depolarization below 2km is from marine air mixed upwards (this is my hunch) from continental air. Just looking at some example profiles and examining the air flow - perhaps the information in Ryoo et al., 2021 and Redemann et al., 2021 on the individual flights is enough.

Figs 5-10: the titles on all of these panels are difficult to read.

References:
J.-M. Ryoo, et al., 2021: A meteorological overview of the ORACLES (ObseRvations of Aerosols above CLouds and their intEractionS) campaign over the southeast Atlantic during 2016-2018. Part 1 - Climatology *Atmos. Chem. Phys.*, **21**, 16689-16707, doi:10.5194/acp-21-16689-2021

K. Pistone, et al., 2021: Exploring the elevated water vapor signal associated with the free-tropospheric biomass burning plume over the southeast Atlantic Ocean. *Atmos. Chem. Phys.*, **21**, p. 9643-9668, doi:10.5194/acp-21-9643-2021

J. Zhang and P. Zuidema, 2021: Sunlight-absorbing aerosol amplifies the seasonal cycle in low cloud fraction over the southeast Atlantic: *Atmos. Chem. Phys.*, **21**, p. 11179-11199, doi:10.5194/acp-21-11179-2021

Kacarab, M., et al. 2020: Biomass Burning Aerosol as a Modulator of Droplet Number in the Southeast Atlantic Region. *Atmos. Chem. Phys.*, **20**, p. 3029-3040, doi:10.5194/acp-20-3029-2020

Haywood et al., 2021: Overview: The CLoud-Aerosol-Radiation Interaction and Forcing: Year-2017 (CLARIFY-2017) measurement campaign, *Atmos. Chem. Phys.*, **21**, p. 1049-1084, doi:10.5194/acp-21-1049-2021

---

## Author Comment (AC1)

September 10 2016 1200 UTC
650 hPa Winds

September 12 2016 1200 UTC
650 hPa Winds

September 14 2016 1200 UTC
650 hPa Winds

September 16 2016 1200 UTC
650 hPa Winds

September 18 2016 1200 UTC
650 hPa Winds

September 20 2016 1200 UTC
650 hPa

September 22 2016 1200 UTC
650 hPa Winds

September 24 2016 1200 UTC
650 Pa

September 26 2016 1200 UTC
650 hPa Winds

---

## Author Response (AR1)

Review of:
Vertical structure of biomass burning aerosol transported over the southeast Atlantic Ocean by:
Harshvardhan et al.
Manuscript doi:10.5194/acp-2021-846

This study examines HSRL2 profiles over the southeast Atlantic and interprets them for insights into aerosol microphysical behavior and transport. This is a valuable objective given the quality of the data, its relevance for satellite data interpretation, and the dominance of smoke above clouds for this region. The work hasn't been carried through as completely as it needs to be for publication, however. Major requests to the authors include the following:

Line numbers refer to the revised, marked up document with tracking, not the original document reviewed or the version without tracking.

- we need more synoptic context for the days forming the data sample. When was the African Easterly Jet-south active? Did mid-latitude disturbances ever constrain the aerosol to be closer to the coast?

Pg. 3,4, lines 91-127. We have added a new sub-section, 2.1, Meteorology, that addresses this question. There is also an additional Fig. 2 that shows 650 hPa winds at 1200 UTC on four days during the IOP. A separate explanation with a 9-panel figure was uploaded to the ACP discussion site (https://acp.copernicus.org/preprints/acp-2021-846/#discussion) as a response to this very question. We think this 4-panel figure will suffice. Moreover, since a true Lagrangian study cannot be made with aircraft speeds being 10-20 times the speed of air parcels, Fig. 4 is the more relevant figure. It shows the predominant route taken by air parcels over a two-week period and justifies our choice of upstream/downstream pairs of grid boxes. Our strategy is similar to the referenced strategy of Shinozuka et al. (2020).

- How do the Lidar profiles correspond with the humidity structure, in particular the extinction? Does this explain the vertical increase in extinction with height? You can use MERRA2 or ERA5 to get the humidity structure, and the Pistone et al. 2021 paper to rationalize the choice.

Pg. 6, lines 210-237. We have added a discussion of the humidity field and its relationship to extinction along with references to Pistone et al. (2021) and Doherty et al. (2022).

- the figure panel titles are often difficult to read and should be improved.

Done.

Recommendation: accept with major revisions.

More specific comments are shown below, some of which are repetitive of each other and/or the major comments above.

Abstract:
Line 15: Should HSRL ->HSRL-2?

Corrected.

Last sentence: it would be worth repeating here that the 'clear' gap is associated with September.

It is mentioned in the body of the manuscript; we do not think it is necessary in the Abstract.

Introduction:
Line 32-33: It would be nice to find a quote from the most recent IPCC AR6 to replace this one.

IPCC AR6 is not a finished product yet; the download states 'Do Not Cite, Quote or Distribute'.

Line 41: the recent paper by Kacarab et al., 2020 would serve to underpin this statement. Overall, it would be good to include some updated references. See the ACP special issue.

Pg. 2, lines 48-50. A sentence referring to observations and modelling of aerosol-cloud interactions and references to Diamond et al. (2018), Kacarab et al. (2020) and Gupta et al. (2021) have been added.

Lines 56-63: how about also highlighting the ability to discern aerosol-cloud contact? This will affect the reflectance of the underlying cloud surfaces.

Pg. 2. Please see above.

Lines 75: the appropriate reference for CLARIFY is Haywood et al., 2021.

Pg. 3, line 83. Corrected.

Line 85: frame => framework.

Pg. 3, line 99. Changed.

Lines 92-101: we need more information on why grid boxes A and C were chosen. How much data fell into these boxes and from which days? What distinguishes the synoptics establishing A-B versus C-D? fig. 3 only shows the monthly-mean fields. See Ryoo et al., 2021.

This question has been answered in the very first response above. Fig. 4 (old Fig.3) is not a 'monthly mean' exactly. It shows the history of trajectories of individual air parcels over a two-week period. The choice of upstream/downstream grid boxes was made by selecting the predominant tracks; a true Lagrangian study is impossible with this setup.

Line 147: I think you just show the effective radius, from this list of possible retrieved variables. Would be good to clarify.

Pg. 5, lines 181-182. Sentence added, "Here we restrict ourselves…".

Lines 154-157: yet this list is missing some of the variables you do show….

Pg. 6, lines 189-193. All four lidar variables are plotted in Figs. 6-9.

Line 160: the dates and amount of data from each contributing flight should be provided for each of the boxes. Figure 2 indicates them, but it is qualitative. it might be helpful to show the 2 by 2 degree boxes on fig. 2. Do the northern pairs correspond to when the African easterly jet- south was active? September is also a month when mid-latitude disturbances to the south constrain aerosol to closer to the coast (Ryoo et al., Zhang and Zuidema 2021) and the anomalous ascent can help mix the aerosol (and humidity) upwards - that might explain why the profiles in C-D show the more linear increase in extinction with height. This would also help explain the increase in the depolarization ration below 2km for D.

Response after next comment.

Line 172-174: interesting, that the northern and southern plumes differ in this fashion. Do you have an idea why? Is the extinction correlated with humidity? You could use the MERRA2 relative humidity profiles to ascertain.

The questions regarding meteorology and humidity posed in the two remarks above have been answered earlier in the Response. The dates and number of profiles are marked on the figures, and also in Table 1.

Fig. 2: explain in caption what the letters within the figure mean. Label latitude lines.

Pg. 16, Fig. 3. The caption has been amended. Latitude/longitude lines are already marked on the Google Earth map.

Figure 3: use the same spatial domain for the top and bottom figures. What do hysplit back trajectories at 2.5 km look like? Do those also connect to land? I ask because the back trajectory duration is so much longer for those in fig 4. This would also help explain if it is marine or continental air with different aerosol properties getting incorporated into the retrievals. Would a case study be helpful?

Pg. 17, Fig. 4. These are the default maps drawn by the open version of HYSPLIT. Fig. 4 is explained in great detail in lines 105-127. Fig. 2 shows the relative position of the five grid boxes. Wind speeds are slower at other altitudes and this result is shown in Fig. 5. On p. 8, line 297, we have stated that the retrieved properties apply to the altitude range from 3-5 km, considered 'the smoke plume' where wind speed and direction do not change appreciably

Fig. 5: It would be helpful to put these time ranges in synoptic context. Visit Ryoo et al., 2021. What was the relative humidity and how does it affect these results?

The questions regarding meteorology and humidity posed above have been answered earlier in the Response.

Fig. 7: it should be possible to discriminate whether the enhanced depolarization below 2km is from marine air mixed upwards (this is my hunch) from continental air. Just looking at some example profiles and examining the air flow - perhaps the information in Ryoo et al., 2021 and Redemann et al., 2021 on the individual flights is enough.

Pg. 7, lines 250-254. Explained – "…mixing with marine and continental particles from regions not affected by biomass burning".

Figs 5-10: the titles on all of these panels are difficult to read.

Figs. 6-12. Labels have been enlarged

References:

J.-M. Ryoo, et al., 2021: A meteorological overview of the ORACLES (ObseRvations of Aerosols above CLouds and their intEractionS) campaign over the southeast Atlantic during 2016-2018. Part 1 - Climatology *Atmos. Chem. Phys.*, **21**, 16689-16707, doi:10.5194/acp-21-16689-2021

K. Pistone, et al., 2021: Exploring the elevated water vapor signal associated with the free-tropospheric biomass burning plume over the southeast Atlantic Ocean. *Atmos. Chem. Phys.*, **21**, p. 9643-9668, doi:10.5194/acp-21-9643-2021

J. Zhang and P. Zuidema, 2021: Sunlight-absorbing aerosol amplifies the seasonal cycle in low cloud fraction over the southeast Atlantic: *Atmos. Chem. Phys.*, **21**, p. 11179-11199, doi:10.5194/acp-21-11179-2021

Kacarab, M., et al. 2020: Biomass Burning Aerosol as a Modulator of Droplet Number in the Southeast Atlantic Region. *Atmos. Chem. Phys.*, **20**, p. 3029-3040, doi:10.5194/acp-20-3029-2020

Haywood et al., 2021: Overview: The CLoud-Aerosol-Radiation Interaction and Forcing: Year-2017 (CLARIFY-2017) measurement campaign, *Atmos. Chem. Phys.*, **21**, p. 1049-1084, doi:10.5194/acp-21-1049-2021

Atmos. Chem. Phys. Discuss., referee comment RC2
https://doi.org/10.5194/acp-2021-846-RC2, 2022

[Figure]

**Comment on acp-2021-846**

Anonymous Referee #2
* * *
Referee comment on "Vertical structure of biomass burning aerosol transported over the southeast Atlantic Ocean" by Harshvardhan Harshvardhan et al., Atmos. Chem. Phys. Discuss., https://doi.org/10.5194/acp-2021-846-RC2, 2022
* * *
Line numbers refer to the revised, marked up document with tracking, not the original document reviewed or the version without tracking.

The paper presents averaged profiles of extinction, backscatter Angstrom exponent, aerosol depolarization and aerosol lidar ratio of biomass burning smoke as measured with an HSRL-2, as well as HSRL-2 retrievals of aerosol submicron fraction and the size of the sub-micron fraction, from the ORACLES project 2016 field deployment in the SE Atlantic. Data are aggregated into two pairs of 2degree x 2degree gridboxes, with one gridbox downwind of the other. An additional gridbox is included south of the biomass burning plume for contrast. The average properties of the aerosol, dominated by biomass burning smoke, are shown to be relatively invariant in the approximately 1day+ of transport from the upwind to the downwind gridbox.

The study is limited in scope but of scientific value and appropriately short. I don't have any significant problems with the analysis presented and think it's a solid and useful piece of evidence that the aerosol intensive properties measured don't vary significantly over a fairly broad region of the SE Atlantic. I do think the paper needs to be more carefully written. As such, I recommend publication after the many smaller issues below are addressed.

Smaller comments:

Abstract, pg 1, lines 20-21: "The fraction of aerosol in the fine mode between 50 and 500 nm remained above 0.95 and the effective radius of this fine mode was 0.16 um from 3 to 5 km in altitude." The wording here could use some work, and it would be better to stick to either nm or um. E.g: "In the 3-5km altitude range, 95% of the aerosol mass (is mass correct?) was in the 50-500nm radius size range, with the aerosol in this size range having an average effective radius of 160nm"

Pg. 1, lines 20-22. Followed reviewer's suggestion and adopted particle radius in micrometers for consistency with the rest of the manuscript. It is not mass but extinction and this has been stated clearly now. "…95% of the aerosol extinction was contributed by particles in the 0.05-0.50 μm radius size range"

Pg. 2, line 28-29: "Aerosols are often considered as the most confounding element in the climate system when simulating future parameters of Earth's climate." This isn't accurate. Aerosols are the most uncertain climate forcer in present day, as stated a few sentences later, but other things introduce much greater uncertainty in future climate. E.g. future GHG emission trajectories, cloud feedbacks and how climate warming might affect natural carbon sources and sinks are greater uncertainties in future climate projections than are aerosols.

Pg. 1, lines 30-31. Changed future to current.

Pg 2, line 38: "overly" to "overylay"

Pg. 2, line 46. Changed.

Pg 2, lines 46-47: "by mostly absorbing the incoming solar radiation" This isn't correct; the single-scatter albedo of these aerosols is ~0.85, so they scatter much more sunlight than they absorb.

Pg.2, line 57. Deleted mostly.

Pg 2: In discussing the aerosol mixing into clouds, there are at least two studies from ORACLES that could be cited: Diamond et al. (2018); https://doi.org/10.5194/acp-18-14623-2018 and Gupta et al. (2021) https://doi.org/10.5194/acp-2021-677

Pg. 2, lines 48-50. A sentence referring to observations and modelling of aerosol-cloud interactions and references to Diamond et al. (2018), Kacarab et al. (2020) and Gupta et al. (2021) have been added.

Pg. 4, line 108: "at the grid box center" Do you mean that arrived within the 2degx2deg gridbox? This reads as if it only includes those data points that exactly fall at the gridbox center. Is this the case?

Pg. 4, line 135. Deleted center.

Pg 4, line 121: I'm not sure what the use of the word "tenuous" here is intended to mean.

Pg. 4, line 148. Reference to tenuous clouds has been removed since we are measuring aerosol only here.

Pg 5, line 153: "Aerosol extinction… the primary measure of aerosol abundance". Aerosol mass or number concentration is the primary measure of aerosol abundance (depending on what question you're trying to answer). Extinction depends on mass, size and relative humidity. The role of relative humidity in determining extinction needs to be acknowledged. Two articles that address the role of humidity in the context of aerosol extinction from the ORACLES campaign and that could be referenced are Pistone et al. (2021; https://doi.org/10.5194/acp-21-9643-2021) and Doherty et al. (2022; https://doi.org/10.5194/acp-22-1-202)

Pg. 6, lines 189, 210-237. Changed aerosol extinction property; added a discussion of the humidity field and its relationship to extinction along with references to Pistone et al. (2021) and Doherty et al. (2022).

Pg. 6, line 171: "essentially constant extinction from just above 2.5 km to 5 km" For box B, the extinction is a bit more variable, so I'd reword this slightly to "relatively constant extinction"

Pg. 6, line 208. Changed as per reviewer's suggestion.

Pg. 6 lines 171-187 The discussion here needs to reference the figures in the appropriate places.

Pg. 6, 7, lines 199-237. I think we have done this correctly. A new Figure 10 showing relative humidity has been added after the lidar plots.

Pg. 6, lines 174-177: "The Lidar Ratio above 3 km … is considerably less and highly variable in Box E and in the lower layers of the aerosol plume in Box D, where the plume most likely has components of continental aerosol typical of the nearby Namibian coast." What is the basis for assuming that the lower part of the plume in Box D is "continental aerosol"? And what does "continental aerosol" mean? Dust? Pollution? A mix of BB smoke, dust and pollution?

Pg. 7, lines 250-254. Explained – "…mixing with marine and continental particles from regions not affected by biomass burning".

Pg. 6, lines 177-179: "The most striking feature of the results are the near constant values of these parameters in the upper two kilometers of the plume over the course of several days as evident from the range of values in the 25-75 percentile shaded grey in Figs. 6 and 7."

The text right before this was discussing the lidar ratio, but the lidar ratio is shown in Figure 8 so it's not clear what "these parameters" is referring to?

Also, this text is a bit ambiguous. It can be read to mean that "these parameters'" mean values don't change much in the upper 2km of the plume, which I don't see to be the case; both the backscatter Angstrom exponent (Figure 6) and (to a lesser degree) aerosol depolarization (Figure 7) change with altitude. Do you mean that at a given altitude these properties are quite invariant?

Pg. 7, line 243. Yes, invariant; explained clearly now, "…is the very small profile-to-profile variability of the intensive lidar parameters…"

Pg. 6, lines 179-181: "This suggests strongly that the original particles near the source of combustion have been coated before they cross the land-ocean boundary and maintain their size over the first few days of transport over the ocean." What is the relationship between a change in aerosol size driven by changes in coating and changes in the lidar ratio, Angstrom exponent and depolarization? How sensitive are each of these parameters to changes in coating and size? Please be more quantitative.

Pg. 7, line 249. Since we are not covering chemical changes to the particles, we have reworded the sentence to state simply that particles are spherical with unknown inhomogeneities within.

Pg. 6, lines 181-182: "The lower portion of the plume containing larger BB aerosol particles is subject to mixing with other particles and is highly variable in nature." Why is the lower part of the plume more subject to mixing with "other particles" than the upper part of the plume? What is the source of these "other particles"?

Pg. 7, lines 251-252. Explained – "…mixing with marine and continental particles from regions not affected by biomass burning"

pg. 7, lines 212-213: "The increase in depolarization (Fig. 7) at these lower levels and a decrease in the Lidar Ratio (Fig. 8) suggest mixing with continental and marine particles." Again, what is meant by "continental" particles needs to be specified.

Pg. 8, line 294. As above.

Pg 7, lines 215-216: "The effective radius is 0.16±0.1 um with little variation throughout the vertical extent of the plume." Is this average +/- standard deviation calculated within given altitude ranges? i.e., what is defined as "the plume"? It would be very helpful in the figure to show what's considered "in the plume", for example by adding horizontal lines to the figures defining the bottom and top of the plume.

Pg. 8, lines 297. The word "plume" has been replaced by "between 3 and 5 km"; +/- 0.01 μm has been removed. The sentence "The effective radius…" is simply a description of Fig. 12.

Pg 7, lines 221-222: "Here again, the effective radius of the submicron fraction of particles, 0.15 μm, is uniform with altitude, and comparable though biased slightly low." This is confusing: First, is it 0.15 or 0.16 um? (a small difference, but confusing nonetheless). Second, "biased slightly low" compared to what? Or are you talking about the results from Sawamura et al.?

Pg. 8, line 303-304. Sawamura et al. obtained 0.15 μm and their value was biased slightly low compared to in situ measurements.

Pg 7, line 236: I don't understand what is meant by the aerosol being "tenuous".

Pg. 9, line 332. Replaced by "…have very low extinction coefficients suggesting low abundance.

---

## Referee Report (RR1)

I am basically satisfied with the revision. I thank the authors for including the additional analysis of the moisture fields, which help deepen the study. I have some additional minor comments listed below. The most major one is that some discussion of how the HSRL-2 derived effective radii compare to what's been reported from the field campaigns, and what it means, would be nice

Line 49: "are in" -> "include"

Line 51: other more recent modeling studies quantifying the semi-direct radiative effect include Mallet et al 2020 ACP and Solmon et al 2021 npc climate and atmospheric science

Line 59-61: sentence a bit vague as written, the 3 studies cited I believe all focus on an increase in cloud cover/LWP by aerosol absorption occurring above the cloud. Aerosol embedded within the cloud layer can indeed reduce cloud cover through raising the temperature and lowering the relative humidity, shown, e.g., in Zhang and Zuidema 2019 ACP using data from ascension island.

Top of page 3: the way it's written is slightly confusing in that the paragraph under '2' suggests data from 3 campaigns will be used, but I think this study just focuses on September 2016, and only oracles data. This doesn't entirely come through.

Line 91: add 'September' after monthly-mean

Line 93: IOPs not defined. You could just say 'deployments', even clearer would be substituting 'for the September 2016 deployment' for 'for all ORACLES IOPs'. A basic description of the AEJ-S would also be helpful.

Line 313: if the authors can find some particle sizes from the campaign literature to cite here that would add interest. Wu et al 2020 ACP show PCASP-derived median diameters of about 230 nm for CLARIFY and report similar values from SAFARI data on their p. 12707. Shinozuka 2020 fig 9 shows UHSAS dry mean diameters of about 200 nm for smoke layers only. They didn't do effective radius unfortunately. Do these 2 studies suggest nevertheless that the HSRL2-derived values may be biased slightly high? The ORACLES September 2016 UHSAS and LDMA size data would be publicly available if the authors wanted to do a quick check.

References:
Mallet, M., F. Solmon, P. Nabat, N. Elguindi, F. Waquet, D. Bouniol, et al, 2020: Direct and semi-direct radiative forcing of biomass burning aerosols over the Southeast Atlantic (SEA)and its sensitivity to absorbing properties: a regional climate modeling study. *Atmos. Chem. Phys.*, **20**, p. 13191-13216, doi:10.5194/acp-20-13191-2020
Solmon, F., Elguindi, N., Mallet, M. et al. West African monsoon precipitation impacted by the South Eastern Atlantic biomass burning aerosol outflow. npj Clim Atmos Sci 4, 54 (2021). https://doi.org/10.1038/s41612-021-00210-w
Wu, H., Taylor, J. W., Szpek, K., Langridge, J. M., Williams, P. I., Flynn, M., et al.: Vertical variability of the properties of highly aged biomass burning aerosol transported over the southeast Atlantic during CLARIFY-2017, Atmos. Chem. Phys., 20, 12697–12719, https://doi.org/10.5194/acp-20-12697-2020, 2020.
Zhang, J. and P. Zuidema, 2019: The diurnal cycle of the smoky marine boundary layer observed during August in the remote southeast Atlantic. Atmos. Chem. Phys., 19, p. 14493-14516, doi:acp-19-14493-2019

---

## Author Response (AR2)

Line numbers refer to the revised, marked up document with tracking, not the original revised document reviewed or the version without tracking.

Reviewer #1

I am basically satisfied with the revision. I thank the authors for including the additional analysis of the moisture fields, which help deepen the study. I have some additional minor comments listed below. The most major one is that some discussion of how the HSRL-2 derived effective radii compare to what's been reported from the field campaigns, and what it means, would be nice

Line 49: "are in" -> "include"

Line 51: other more recent modeling studies quantifying the semi-direct radiative effect include Mallet et al 2020 ACP and Solmon et al 2021 npc climate and atmospheric science

Lines 43-44. 'include' and additional reference.

Line 59-61: sentence a bit vague as written, the 3 studies cited I believe all focus on an increase in cloud cover/LWP by aerosol absorption occurring above the cloud. Aerosol embedded within the cloud layer can indeed reduce cloud cover through raising the temperature and lowering the relative humidity, shown, e.g., in Zhang and Zuidema 2019 ACP using data from ascension island.

Lines 56-57. Sentence added with reference to Zhang and Zuidema (2019).

Top of page 3: the way it's written is slightly confusing in that the paragraph under '2' suggests data from 3 campaigns will be used, but I think this study just focuses on September 2016, and only oracles data. This doesn't entirely come through.

Lines 84-85. Clarified by adding 'the deployment covered in this study'.

Line 91: add 'September' after monthly-mean

Line 88. Done.

Line 93: IOPs not defined. You could just say 'deployments', even clearer would be substituting 'for the September 2016 deployment' for 'for all ORACLES IOPs'. A basic description of the AEJ-S would also be helpful.

Lines 90-91. 'Deployments' instead of IOP and a description of AEJ-S.

Line 313: if the authors can find some particle sizes from the campaign literature to cite here that would add interest. Wu et al 2020 ACP show PCASP-derived median diameters of about 230 nm for CLARIFY and report similar values from SAFARI data on their p. 12707. Shinozuka 2020 fig 9 shows UHSAS dry mean diameters of about 200 nm for smoke layers only. They didn't do effective radius unfortunately. Do these 2 studies suggest nevertheless that the HSRL2-derived values may be biased

slightly high? The ORACLES September 2016 UHSAS and LDMA size data would be publicly available if the authors wanted to do a quick check.

Lines 267-283. A paragraph has been added along with several references regarding particle size. The HSRL-2 retrievals of SMF effective radii are discussed in the context of previous measurements and retrievals. A comparison of HSRL-2 derived particle size with in situ measurements is not straightforward since the retrieved sizes are for ambient particles whereas airborne in situ measurements are for dry aerosol.

References:
Mallet, M., F. Solmon, P. Nabat, N. Elguindi, F. Waquet, D. Bouniol, et al, 2020: Direct and semi-direct radiative forcing of biomass burning aerosols over the Southeast Atlantic (SEA)and its sensitivity to absorbing properties: a regional climate modeling study. *Atmos. Chem. Phys.*, **20**, p. 13191-13216, doi:10.5194/acp-20-13191-2020
Solmon, F., Elguindi, N., Mallet, M. et al. West African monsoon precipitation impacted by the South Eastern Atlantic biomass burning aerosol outflow. npj Clim Atmos Sci 4, 54 (2021). https://doi.org/10.1038/s41612-021-00210-w
Wu, H., Taylor, J. W., Szpek, K., Langridge, J. M., Williams, P. I., Flynn, M., et al.: Vertical variability of the properties of highly aged biomass burning aerosol transported over the southeast Atlantic during CLARIFY-2017, Atmos. Chem. Phys., 20, 12697–12719, https://doi.org/10.5194/acp-20-12697-2020, 2020.
Zhang, J. and P. Zuidema, 2019: The diurnal cycle of the smoky marine boundary layer

Reviewer #2

Re-review by Reviewer #2 of: Vertical structure of biomass burning aerosol transported over the southeast Atlantic Ocean, H. Harshvardhan et al.

Most of the comments in my original review have been addressed. With just a couple small additional changes, I think the paper should be published.

Referring to line numbers in acp-2021-846-ATC1.pdf, which shows the tracked changes to the originally submitted paper:

Lines 56-58: This text still is misleading as it omits the DRF through light scattering. It reads: "It exerts a direct radiative forcing (DRF) on the atmosphere by absorbing the incoming solar radiation along with the radiation reflected by the underlying cloud surface (Chand et al., 2009; Meyer et al., 2013; Zhang et al., 2016)." The DRF is mostly through scattering sunlight, and I'm not sure why there's the focus only on atmospheric absorption here. The atmospheric absorption component can be highlighted while noting by editing to, e.g,: "It exerts a direct radiative forcing (DRF) by scattering and absorbing sunlight in the atmosphere; when underlying clouds are present, these aerosols absorb the incoming solar radiation along with the radiation reflected by the underlying cloud surface."

Lines 50-51. Sentence has been modified to read "…(DRF) by scattering and absorbing solar radiation in the atmosphere; when clouds are present, these aerosols absorb incoming solar radiation…"

Line 201: The text states that Box E is "not significantly influenced by the BB aerosol", but I'm not sure what the basis is for this assertion. There is clearly an elevated layer of extinction at about the same altitudes as in the Boxes A-D, and with similarly aerosol characteristics (Figures 6-9). More to the point,

the ORACLES flights through the area prescribed by Box E did often measure smoke aerosol, albeit at lower concentrations than further north. Unless the authors have evidence otherwise, I think it can be stated that Box E coincides with the southern edge of the region influenced by the African biomass burning plume – but not that it is "not significantly influenced" by the BB aerosol. In fact, I'm not sure why it's important to include Box E, though doing so isn't a problem. As such, I think a minor edit to the above sentence is sufficient.

Line 182. Box E has been identified as being at the southern edge of the region. We would like to keep Box E.

---

## Author Response (AR3)

Line numbers refer to the latest revised, marked up document with tracking, not the earlier revised document reviewed or the version without tracking.

Editor's comment

30 May 2022
Editor decision: Publish subject to minor revisions (review by editor)
by Nikos Hatzianastassiou
**Comments to the author**:
Dear authors,

thank you for the revised manuscript. The reviewer rightfully pointed out that there are data available to test the effective radii of the sub-micron aerosols, and in response to this comment, you added text (lines 267-283 in the revised manuscript). Notably, however, you only cite size data from CLARIFY, for which all observations were near Ascension Island, and retrievals of aerosol size from the 2016 ORACLES mission, using HSRL-2 and RSP remote sensing instruments. Indeed, the sizes from both of these studies are consistent with the sizes you use in the analysis. Nevertheless, you didn't include in the discussion the results shown by Shinozuka et al. (2020), which are from in-situ measurements of sub-micron aerosol size during ORACLES and show larger aerosol sizes. Not including these data in the discussion gives the impression that you are avoiding them because they don't agree as well with the sizes used in your study. So, please make reference to them as well.

Best regards

Lines 267-283. As requested, text has been added to compare our results with Shinozuka et al. (2020) who reported on airborne aerosol sizes measured during ORACLES by an Ultra-High-Sensitivity-Aerosol Spectrometer (UHSAS) deployed on the NASA P-3 aircraft. The dry aerosol reported in the publication is actually smaller, not larger. Figure 9a of Shinozuka et al. (2020) shows dry volumetric mean diameter of 0.2 µm for aerosol in the 3-6 km altitude range so the dry volumetric mean radius is half, around 0.1 µm. Based on the size distributions derived from the UHSAS and remote sensing instruments reported by Xu et al. (2021), this translates to a dry effective radius of about 0.09-0.10 µm. Shinozuka et al. (2020) mentioned that the UHSAS sizes had to be adjusted to account for significant under-sizing of the particles. The explanation for the discrepancy in particle size derived from the UHSAS and HSRL-2 retrievals is that the UHSAS measurements are for dry aerosol whereas HSRL-retrieved size is for ambient aerosol which tends to be 15-20% larger at 70-80% RH.

Lines 370-372. A reference under review has been updated to the published version.

---

## Author Response (AR4)

For now, we will proceed with your manuscript as submitted. However, please adjust your manuscript files before your next file upload (next round of revision or after acceptance) considering the following requirements:

Please also include the copyright statement (© Google Earth) for figure 3 in the caption.

The copyright statement has been added to the figure caption.